# Sequestration of Mercury in Soils under Scots Pine and Silver Fir Stands Located in the Proximity to a Roadway

**DOI:** 10.3390/ijerph18094569

**Published:** 2021-04-26

**Authors:** Piotr Gruba, Mateusz Kania, Dawid Kupka, Marcin Pietrzykowski

**Affiliations:** Department of Forest Ecology and Sylviculture, Faculty of Forestry, University of Agriculture in Kraków, 31-425 Kraków, Poland; matikania@o2.pl (M.K.); dawid.kupka@student.urk.edu.pl (D.K.); m.pietrzykowski@urk.edu.pl (M.P.)

**Keywords:** *Abies alba*, forest soils, Hg:C ratio, mercury, *Pinus sylvestris*, road pollution

## Abstract

Forest soils are the main source of mercury (Hg) in stream water. Stocks of Hg in forest soils are related to several factors, including forest species composition. In this study, the potential source of Hg pollution was a relatively new roadway traversing forested areas. We compared Hg accumulation in soils of two coniferous species: Scots pine (*Pinus sylvestris* L.) and silver fir (*Abies alba* Mill.). The experimental plots were located near the S7 expressway in Central Poland. The stands differed in the length of time they had been exposed to Hg, because different parts of the roadway were built and opened to traffic at different times. We analyzed 480 soil samples from organic horizons (O) and the top 10 cm of mineral soil (A) sampled from six plots. The overall average Hg concentrations (irrespective of forest stand, *n* = 240) was 0.225 mg kg^−1^ in the O horizons and 0.075 mg kg^−1^ in the mineral horizons. The Hg concentration in the O horizons was more than three times greater in fir stands than that in pine stands. The average Hg:C ratios in the O and A horizons were 1.0 and 2.3 mg Hg kg^−1^ C, respectively. Our data does not clearly show the effect of road on Hg accumulation near the road. The concentrations of Hg in investigated soils adjacent to the roadway were only slightly higher than ranges reported for unpolluted areas, and no clearly affected by the vicinity of roadway. In contrast to the other reports, our data indicate a significant impact of tree species on Hg concentrations in both the O and A horizons. Moreover, the average Hg:C ratio was strongly dependent on the tree species.

## 1. Introduction

Mercury (Hg) in soil originates from atmospheric deposition, and the Hg pools in soil depend largely on the vegetation type, particularly in forest ecosystems [1,2,3]. Retention of Hg in forest soils is attributed to the affinity of Hg for soil organic matter (SOM) [2,4,5,6,7]. Soil Hg compounds are bound to organic matter [4,8] and form complexes with thiol groups (R-S-H) [2] in the SOM. The content of organic matter and particle size composition are spatially variable soil characteristics, thus the degree of Hg accumulation is not constant in any given area, but varies depending on local heterogeneity of organic matter, soil texture, and elevation [3]. Given that forest areas are one of the main sources of Hg in stream water [9,10], understanding how Hg accumulates in, and mobilizes from, forest soils is important to understand how atmospheric Hg ultimately affects aquatic ecosystems.

Road traffic is a potential source of Hg. Several authors have suggested that traffic emissions and other related anthropogenic activities are potential sources of metal contamination (for example, via roadway dust). However, few studies have focused on Hg pollution near roads [11,12]. According to Jiang et al. [13], coal and oil combustion are the most important sources of global atmospheric Hg, particularly in China, where about 75% of Hg emissions were related to the combustion of fossil fuels. According to Petroleum statistics [14], some kinds of gasoline contain 0.2–3.3 ng g^–1^ of Hg. Pirrone et al. [15] reported that the global Hg emission from petrol and diesel oil combustion is 378 kg per year. Traffic emission was regarded as the most important anthropogenic metal (including Hg) source by Liu et al. [16] and Liang et al. [17]. Also Ozaki et al. [18] found increased levels of Hg in roadside dust and soils. Research sites are strongly affected by the presence of automobiles, which is believed to be responsible for the higher concentrations.

Soil C, particle size composition and altitude are most important factors affecting Hg content in forest soils [3,19]. The influence of forest type is also important, because the quality and content of SOM in forest soils are strongly affected by the forest tree species [3]. Thus, it is expected that forest species composition will affect Hg accumulation in soils. Previous studies have detected clear differences in the ability of certain tree species (particularly coniferous vs. broadleaf) to intercept atmospheric deposition [19,20,21,22,23,24]. Given that C is a major variable affecting the Hg concentration in soil, the effects of other variables (e.g., distance from the pollution sources or the effect of tree species) can be difficult to detect. Thus, Hg to carbon (C) or sulfur (S) ratios are sometimes used as a reliable standardization method [3,24].

The intensive industrial and urban development in Poland has led to the deforestation of large forest areas to accommodate new roadways.

In this study, we tested the hypothesis that two coniferous species with different scavenging abilities, Scots pine (*Pinus sylvestris* L.) and silver fir (*Abies alba* Mill.), significantly affect Hg sequestration close to a roadway. Given the importance of SOM for Hg sequestration, we also analyzed Hg to C and Hg to S stoichiometry in detail.

## 2. Materials and Methods

### 2.1. Study Site

The six study plots were located near roadway S7, between the cities of Skarzysko-Kamienna and Kielce, central Poland. These plots were established in Scots pine (*Pinus sylvestris* L.) and Silver fir (*Abies alba* Mill.) forest stands, on soils developed from three kinds of parent materials (Table 1). According to the Geological Map of Poland [25], the local soils are formed of Quaternary sands that originated from weathering sandstones, Triassic sandstones formed in the lower Triassic, and Cambrian sandstones. The soils were classified as Cambisols and Podzols [26]. The terrain at the study site was relatively simple, and the elevation varied from 253 to 340 m above sea level. Forest stands with fir were unevenly aged with a multistoried texture, whereas Scots pine forests were evenly aged and one-storied (Table 1).

The investigation sites were relatively close (5–16 km) to the reference area investigated previously by Gruba et al. [27]. The area was established on similar type of parent material (see Figure 1b and Table 1). Due to long distance to the nearest road or any potential source of Hg, the area was considered to be relatively unpolluted.

At each plot the sampling points were located according to the scheme shown in Figure 1. Soil samples were taken from various distances from the road, from the organic horizon (O) and first 10 cm of the mineral (A) horizon.

### 2.2. Laboratory Analyses

Prior to analysis, soil samples were air-dried at room temperature and then sieved through 2 mm mesh. We used 2 mm-sieved samples for measurement of the particle size composition by laser diffractor (Fritsch Analysette 22, Fritsch, Idar-Oberstein, Germany). The pH was measured potentiometrically with a combination electrode in a suspension with distilled water (1:5, *w*/*v*) after 24 h of equilibration. Cation exchange capacity (CEC) was estimated a sum of exchangeable cations (BC = Ca^2+^ + Mg^2+^ + K^+^ + Na^+^) and total acidity (TA). Exchangeable cations were extracted with 1 mol L^−1^ NH_4_Cl and determined by an ICP (ICP-OES Thermo iCAP 6500 DUO, Thermo Fisher Scientific, Cambridge, UK). TA was measured after extraction of soilwith 1 mol L^−1^ (CH_3_COO)_2_Ca and determined using automatic titrator, Mettler Toledo, Inc. to pH 8.2 with 0.1 mol L^−1^ NaOH. Soil subsamples were ground into a fine powder using a ball mill (Fritsch) to improve homogeneity. These fine samples were used to measure the total Hg content with a direct Hg analyzer DMA-80 (Milestone, Sorisole, Italy). To assure the quality control (QC) of Hg measurements, the analysis included standard European Reference Material (ERM) N_o_ CC141 (Loamy Soil), with certified value Hg = 0.083 mg kg^−1^, uncertainty 0.017 mg kg^−1^, recovery between 80 and 120%. Analysis of ERM was performed at the beginning and the end of each experimental series. ERM and each sample were analysed in two replicates, with permissible difference between the measurements <10%. The limit of quantitation was 0.003 mg kg^−1^. The fine subsamples were also used to measure the contents of soil carbon (C) and S with a LECO CNS TrueMac analyzer (Leco, St. Joseph, MI, USA). Since all soils were carbonate-free we assumed that C equals organic carbon.

In order to estimate the bulk density, a separate samples were taken. A 20 cm × 20 cm frame was used to the sampling of organic (O) horizon. The thickness of the organic horizon was measured during sampling. Samples from mineral soil horizons were sampled with metal cores (250 cm^3^). Samples were weighed after drying at 105 °C for 24 h.

Soil Hg pools were calculated per 1 square meter as the sum of Hg pools in the O and A horizons, based on measured Hg concentrations and bulk densities for each horizon.

### 2.3. Statistical Analyses

The dataset was divided into groups by the plot, tree species, or distance from the roadway edge and descriptive statistics were calculated. To analyze differences in the Hg, and C contents, as well as the Hg:C ratio, and other soil properties measured the different groups, Z test was performed to test statistical differences between pairs or Kruskal-Wallis was performed to compare multiple independent groups. Significance was accepted at *p* < 0.05. Relationships between variables were investigated by performing correlation and regression analyses. We used the Pearson product–moment correlation coefficient (r). Multiple regression was used to evaluate the relationships between Hg and independent variables (i.e., soil properties and forest stands characteristics). The Hg and C content data were log-transformed to approach data normal distribution. All statistical analyses were performed using STATISTICA 12 software (StatSoft, Tulsa, OK, USA).

## 3. Results

The properties of the investigated soils of the fir and pine stands differed considerably. Soils of pine stands, derived from glacio-fluvial deposits, were sandy (average contents: sand 78%, silt 19% and clay 3%). Soils of fir stands, developed on weathered sandstones, were siltic (average contents: silt 57%, sand 35%, clay 7%). Soil of fir stands also had more organic carbon accumulated in both the O and mineral A horizons (Table 2), resulting in higher C contents in soils of fir (6.52 kg C m^−2^) than in soils of pine (4.32 kg C m^−2^). Note that at the investigated depth, the majority of C (~60%) was allocated to the mineral A horizon (10 cm). The S content was significantly higher in soils of fir stands (both in the O and A horizons) than in soils of pine stands. Both fir and pine soils were strongly acidic (Table 2), but soils of pine stands had significantly higher pH values. Considering vertical differences in pH, the average pH values of the O and A horizons of pine stands were not significantly different, whereas the O horizons had significantly higher pH values than those of the mineral (A) horizons in soils of fir stands (Table 2).

The overall average Hg concentrations (irrespective of forest stand, *n* = 240) were 0.225 mg kg^−1^ in the O horizons and 0.075 mg kg^−1^ in the mineral horizons. The Hg content in soil differed significantly between the fir and pine stands. The average Hg concentration in the O horizons was more than three times greater in fir stands than that in pine stands (Table 2). In the A horizons, the difference was about four-fold. In the O horizons, the Hg concentration was highest at plot 4 (fir stand, Hg = 0.781 mg kg^−1^) and lowest at plot 5 (pine stand, Hg = 0.017 mg kg^−1^). In the A horizons, the highest Hg concentration was at plot 4 (fir stand, Hg = 0.378 mg kg^−1^) and the lowest was at plot 2 (pine stand, Hg = 0.01 mg kg^−1^). The Hg concentration data for all plots were compared using a Kruskal-Wallis test. The plots could be ranked, from highest average Hg concentration to lowest, as follows: O horizons: plot 4 (0.519 mg kg^−1^) ~ plot 3 (0.466 mg kg^−1^) > plot 1 (0.388 mg kg^−1^) > plot 2 (0.211 mg kg^−1^) > plot 5 (0.142 mg kg^−1^) > plot 6 (0.083 mg kg^−1^); A horizons: plot 4 (0.153 mg kg^−1^) ~ plot 3 (0.135 mg kg^−1^) > plot 1 (0.061 mg kg^−1^) ~ plot 6 (0.043 mg kg^−1^) ~ plot 2 (0.041 mg kg^−1^) > plot 5 (0.023 mg kg^−1^).

The average Hg stock in soils (i.e., sum of stocks in the O and A horizons) was 10.1 ( ± 6.1) mg m^−2^. The average Hg stock was three times higher in fir stands (15.3 ± 5.7 mg m^−2^) than that in pine stands (5.4 ± 1.9 mg m^−2^).

To investigate the effect of duration of exposure to the roadway on soil Hg concentration we compared the data from plots near the roadway opened 8 years ago (four plots, 160 sampling points) to the data from plots with about 35 years of exposure (two plots, 80 sampling points). For fir stands Hg stocks in the O + A horizons were significantly higher near the roadway opened 35 years ago than in soil near the roadway opened 8 years ago, whereas for pine stands there was no significant difference (Figure 2).

The Hg concentration in soil was significantly (*p* < 0.05) positively correlated with C content (r = 0.48 and 68 in the O and A horizons, respectively). To standardize the concentrations of Hg samples with variable C content, we calculated the Hg:C ratio. The average Hg:C ratio in the O horizons (1.0 mg Hg kg^−1^ of C) was about half that in mineral soil (2.3 mg Hg kg^−1^ of C). Comparing soils of fir and pine stands, the Hg:C ratio in the O horizons of pine stands was half of that in the O horizons under fir stands. In the A horizons, the Hg:C ratio was significantly higher than that in the O horizons, but the difference in values between soils of fir and pine stands was similar. Similar trends were detected for the Hg:S ratio; i.e., the Hg:S ratio in soils of fir stands was significantly higher than under that pine stands for both the O and A horizons (Table 2). The relationships between C and Hg concentrations were linear. In the O horizons, the gradients of the relationships were similar for soils of fir and pine stands, but the intercept was clearly higher for soil of fir stands than for soil of pine stands (Figure 3a). In contrast, for the A horizons, the intercepts were similar but the gradient was steeper for soil of fir stands than for soil of pine stands (Figure 3b).

We also investigated the effect of very close proximity (50 m) to the roadway on Hg concentrations in forest soils. In the O horizons, the Hg concentration was only decreased in close vicinity (2 m) to the forest edge in fir stands (Figure 4a). We detected no clear patterns in the relationships between Hg concentration and distance to the road in the O and A horizons of pine and fir stands (Figure 4a,b). We then plotted the Hg:C ratio as a function of distance from the road (Figure 5). In the A horizons of fir stands, the Hg:C ratio was significantly lower in the plots 2, 7, and 12 m from the roadway than in plots further from the roadway (Figure 5b). In contrast, under pine stands, the Hg/C ratio slightly increased near the roadway. Such changes were not detected in the O horizons (Figure 5a).

Multiple regression analysis revealed that in both the O and A horizons, three variables significantly contributed to the model describing the Hg concentration, namely: the contents of C and S and the species (fir or pine), which collectively explained 76% and 72% of Hg concentrations in the O and A horizons, respectively. The C content alone explained 28% and 43% of variation in Hg concentrations in the O and A horizons, respectively. In the O horizons, the S concentration explained an additional 21% of the variation in Hg concentration, whereas the variable “tree species” explained 27% of the variation in Hg concentration. In the A horizons, the S concentration explained 14% of the variation in Hg concentration, whereas “tree species” explained 15%.

## 4. Discussion

Despite close vicinity to the roadway, the soil concentrations of Hg (0.225 mg kg^−1^ in the O horizons and 0.075 mg kg^−1^ in the A horizons) lay within the ranges reported for weakly polluted areas. In the literature, more available data exist for the mineral A horizons than for the O horizons. The FOREGS Geochemical Atlas [28] reported 0.011 mg Hg kg^−1^ for the majority of Poland’s territory. For the reference area located ca. 4.75 km from the highway in the center of the investigated forest area, Gruba et al. [27] reported 0.076 mg Hg kg^−1^ in the A horizons (mixed fir-beech-pine stands). Gruba et al. [3] analyzed a large data set from throughout Poland, and detected average values of 0.13 mg Hg kg^−1^ and 0.04 mg Hg kg^−1^ in the O and A horizons, respectively. However, most of those data were collected from soil of pine stands.

A broad spectrum of soil parameters can affect the ability of soils to sequester Hg, such as the contents of C and S, texture, and forest type. The concentrations of total Hg in soil varied between 0.01 and 0.781 mg kg^−1^ in this study. The major variable explaining variations in Hg concentrations was C content, which reflects the SOM content. In soil, Hg has a strong affinity for SOM [2,4]. As shown in Figure 3, there was a linear relationship between soil C content and total Hg content. Compared with data in other reports [5,27], our results demonstrated quite good linear relationship between Hg and C contents, whilstin several studies, the original data were scattered but the relationships become more linear when both variables were log-transformed [5,27].

The Hg content in soil can be standardized by calculating the Hg:C or Hg:S ratio. In the soils studied here, the average Hg:C ratios in the O and A horizons were 1.0 and 2.3 mg Hg kg^−1^ C, respectively. Based on the literature, it can be assumed that Hg:C ratios depend on the decomposition degree of SOM and continuously increase with soil depth. Gruba et al. [27] reported that the Hg:C ratio tends to increase with the soil depth. They found 2.8 and 4.4 mg Hg kg^−1^ C in the topsoil and subsoil horizons, respectively. Moreover, Navratil et al. [7] observed a clear increase in the Hg:C ratio with increasing depth. For a set of forest soils in Poland, Gruba et al. [3] found that Hg:C ratios increased with depth: i.e., 0.7, 1.7, 2.7, and 5.3 mg Hg kg^−1^ C in the O and mineral 0–10 cm, 10–40 cm and 40–100 cm layers, respectively.

There is very little information about the effect of roadways on the accumulation of Hg in soils. In Poland, the highest concentrations of Hg have been detected near mining sites (such as black coal, copper, and other metal ore mines in Silesia region) or smelters [29]. Effects of road is usually attributed to chemical composition of road dust [11,12], which contribute significantly to global Hg pollution [15]. Our data does not clearly show the effect of road on Hg accumulation near the road. However, the comparisons (see Figure 4 and Figure 5) with the average concentration of Hg from the reference area [27] or with average values from Polish forest soils [3] suggested that Hg accumulation can be enhanced near the roadway, but only in soil of fir stands. Also the comparison shown in Figure 2 imply that there is no significant effect of exposure duration (8 or 35 years) on accumulation of Hg, but rather the effect of tree species.

Our data indicate a strong impact of the tree species on Hg concentrations in both O and mineral A horizons. Previous studies [19,23] have concluded that the effect of the dominant tree species is significant only for the O horizon i.e., the soil Hg content in the organic horizon, but not the mineral horizons, is affected by the dominant tree species (pine, spruce, and fir) [3]. However, our results are consistent with those of Navrátil et al. [24], who found higher Hg concentrations at all mineral soil depths under beech stands than under spruce stands. A scavenging effect of vegetation on Hg has been suggested by several authors, that the enhanced deposition under coniferous forest is probably related to the large area of needles year-round to capture Hg from the atmosphere [19,22]. However, differences in forest species are usually combined with differences in orographic precipitation along an elevational gradient. In our study, the elevation of the fir and pine stands was roughly the same, so it was possible to detect differences in their scavenging effects.

The average Hg:C ratios were also found to be strongly dependent on tree species. The trends illustrated in Figure 3 suggest that organic matter of fir stands has more Hg per unit of C. The gradients of the relationships, which reflect the effectiveness of SOM to bind Hg, were similar in soils from pine and fir stands in the O horizons but different in the A horizons. In the A horizons, Hg entered more effectively the soils of fir stands than soils of pine stands. This suggested that the decomposition process enhanced the ability of fir SOM to bind Hg. Considering the effect of tree species on the Hg:C ratio, Navratil et al. [7] found that the Hg:C ratio was higher in soils of spruce stands than in soils of beech stands. Obrist et al. [30] showed that the Hg:C ratios were higher in soils from coniferous stands than in soils of deciduous stands, despite of soils with similar Hg concentrations. In addition, Gruba et al. [3] found that soils of fir stands had a higher level of Hg saturation than soils in stands of other species did, such as pine and oak.

Similar to the Hg:C ratio, the Hg:S ratio appeared to be species dependent, i.e., the Hg:S ratios were significantly higher in soils of fir stands than in soils of pine stands. The data presented in Figure 5 revealed an interesting decline in the Hg:C ratio in the A horizons of soils of fir stands very close (up to 12 m) to the road edge. This decline was followed by a decrease in S content (but not organic C content), implying that soil under fir stands close to the road contained fewer thiol groups able to bind Hg [1,2]. We hypothesize that this is an effect of enhanced decomposition of SOM in this type of stand because of its microclimate. Fir is a shade-tolerant species, and soils of fir stands receive little solar energy. New roads through fir stands affect the local microclimate more than do new roads through pine stands.

In the case of fir stands, changes in forest stand density (e.g., thinning), lead to increase in the decomposition of SOM, and, in the effect, possibly result in the mobilization of Hg accumulated in the organic horizons [31], in higher amounts than in pine stands. The increased transport of Hg in stream waters from forest stands were recognized as a joint result of site preparation and logging by Eklöf et al. [32]. Decaying organic materials also cause a significant increase in Hg leaching during the first several years after harvest [33].

## 5. Conclusions

The concentrations of Hg in soils adjacent to the roadway were slightly higher than ranges reported for unpolluted areas. The concentrations of Hg were not increased in very close proximity (up to 50 m) to the road. Moreover, compared with the average concentration of Hg from the reference area or with average values from Polish forest soils, Hg accumulation was similar or only slightly enhanced near the road. Thus, our results do not confirmed a clear effect of road on soil pollution with Hg. In contrast to other reports, our study detected a strong effect of tree species on Hg concentrations in both the O and mineral A horizons. Although both pine and fir are coniferous species they have different abilities to sequester Hg. The major variable explaining variations in soil Hg concentrations was soil C content. In addition, the average Hg:C ratio was strongly dependent on tree species, i.e., organic matter of fir stands had more Hg per unit of C. In fir stands, a change in forest stand density (e.g., thinning or clear-cutting) can increase the decomposition rate of SOM. This may result in greater mobilization of Hg pools accumulated in the organic horizons than in pine stands. The result of this study should not be interpreted as downgrading of roads as source of Hg pollution. We suggest that the assessment of Hg pollution should take into account variables governing the concentrations of Hg in soils, particularly the content of SOM.

## Figures and Tables

**Figure 1 ijerph-18-04569-f001:**
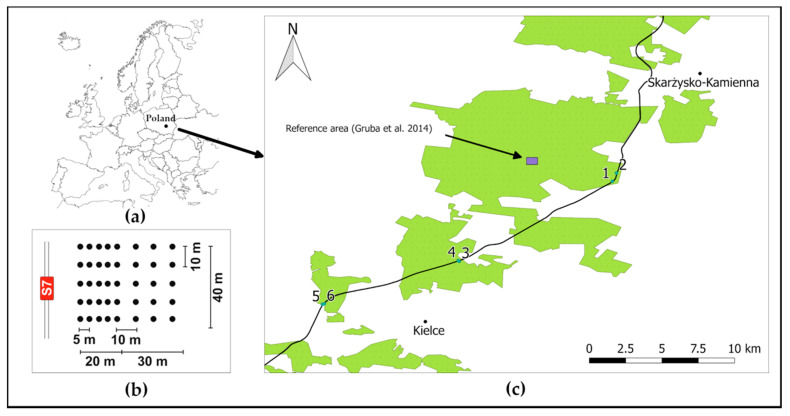
Location of the study sites and the reference area: (**a**) Location of Poland in Europe; (**b**) sampling scheme in each plot; (**c**) locations of the study plots near the S7 roadway and the reference area (blue rectangle).

**Figure 2 ijerph-18-04569-f002:**
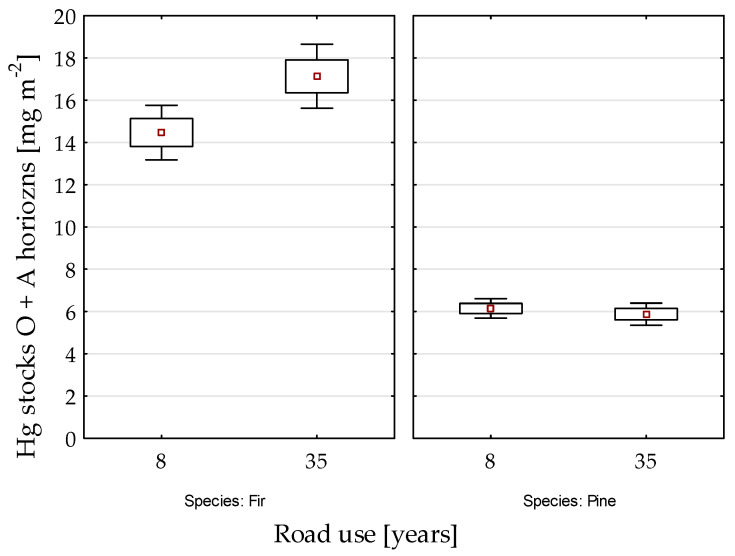
Comparison of Hg stocks in soils (sum of the Hg stocks in the O and A horizons) near roadways opened to traffic 8 and 35 years ago.

**Figure 3 ijerph-18-04569-f003:**
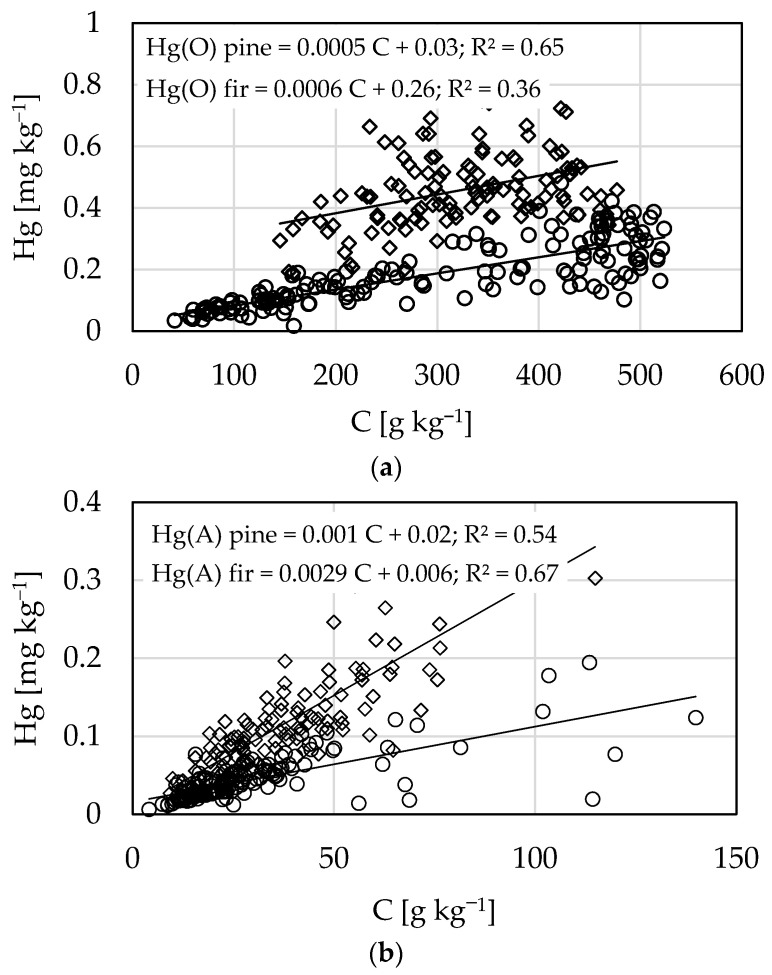
Relationships between mercury (Hg) and carbon (C) contents in soil of pine (○) and fir (◊) stands in the O (**a**) and A (**b**) horizons.

**Figure 4 ijerph-18-04569-f004:**
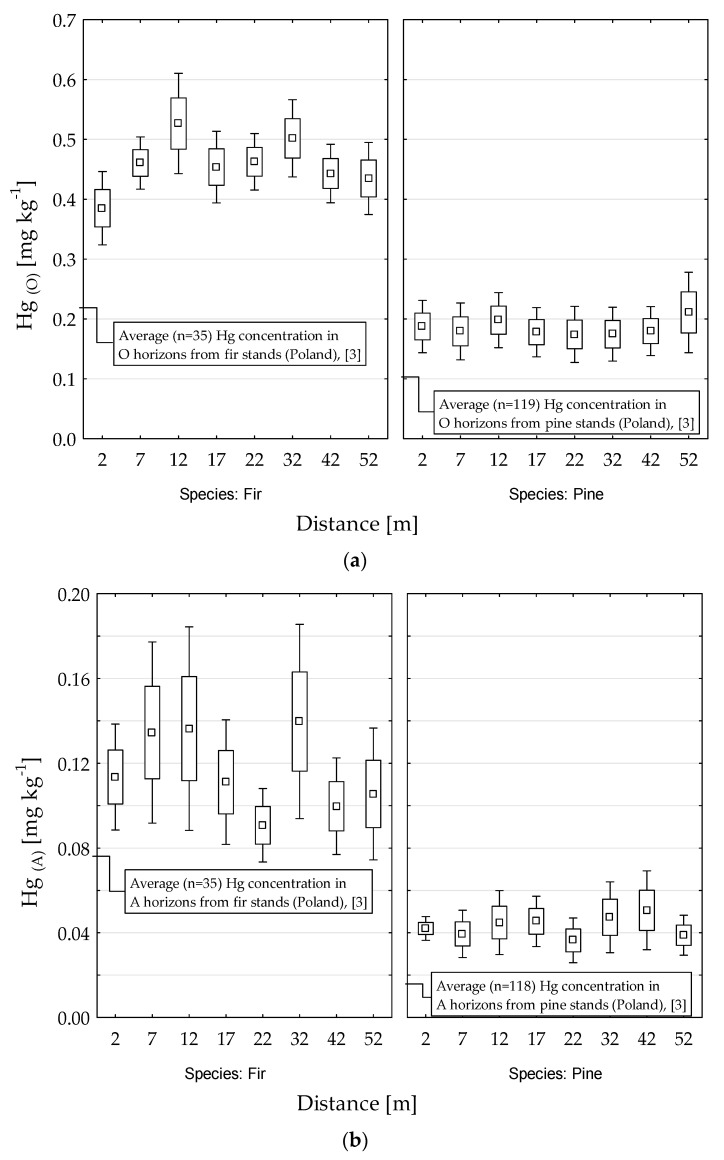
Relationship between mercury (Hg) concentration in the O (**a**) and A (**b**) horizons in soils of fir and pine stands and distance from the roadway.

**Figure 5 ijerph-18-04569-f005:**
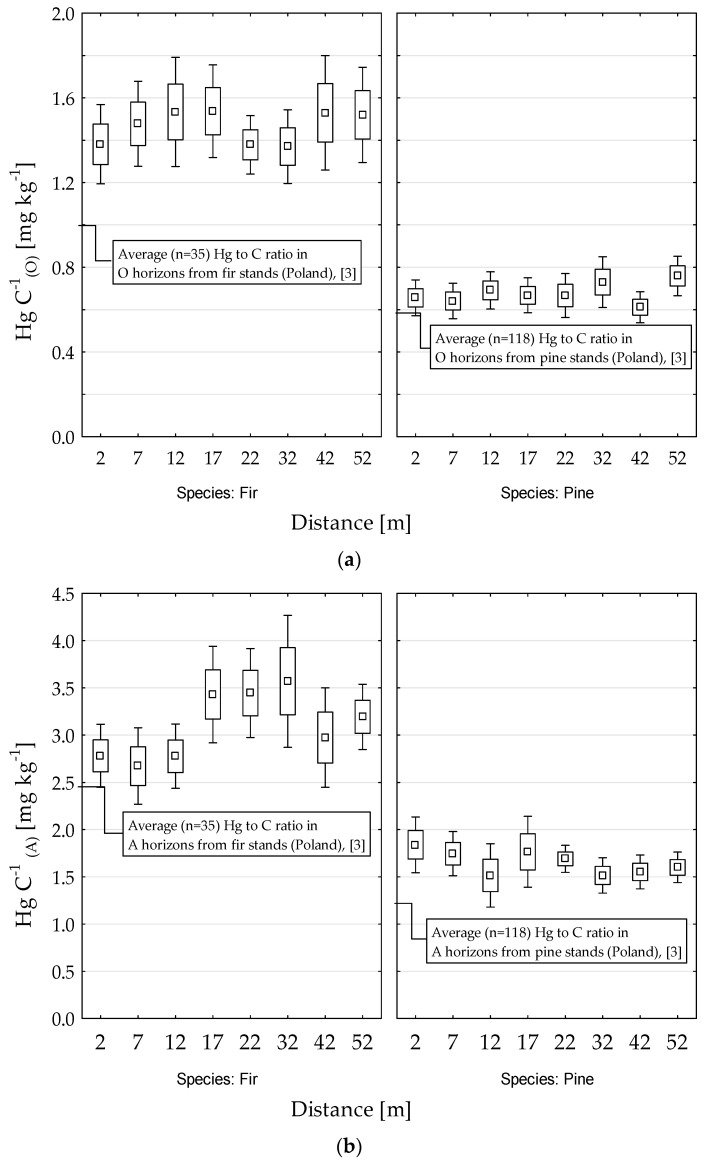
Relationship between mercury to carbon (Hg:C) ratio in the O (**a**) and A (**b**) horizons in soils of fir and pine stands and distance from the roadway.

**Table 1 ijerph-18-04569-t001:** Characteristics of six study plots and reference area.

Plot Number	Geographical Coordinates	Parent Material	Forest Stand Species and Age (Years)	Year S7 Expressway Opened
1	N 51 00.659 E 20 48.551	Triassic sandstones	Silver fir (113) *Abies alba* Mill.	2011
2	N 51 01.137 E 20 48.810	Triassic sandstones	Scots pine (53) *Pinus sylvestris* L.	2011
3	N 50 55.984 E 20 39.586	Cambrian sandstones	Silver fir (93) *Abies alba* Mill.	1984
4	N 50 56.024 E 20 39.519	Cambrian sandstones	Silver fir (94) *Abies alba* Mill.	2011
5	N 50 53.471 E 20 31.560	Quaternary sands	Scots pine (88) *Pinus sylvestris* L.	2011
6	N 50 53.500 E 20 31.677	Quaternary sands	Scots pine (57) *Pinus sylvestris* L.	1984
Reference area [27]	N 51 01 42.92 E 20 43 32.97	Triassic sandstones	Silver fir *Abies alba* Mill. European beech *Fagus sylvativa* L.	**-**

**Table 2 ijerph-18-04569-t002:** Comparison of selected properties of soils of fir and pine stands (mean ± standard deviation). All paired averages from fir and pine stand are significantly different (Z test, *p* < 0.001).

Dominant Tree Species	*n*	C	S	pH_H2O_	Hg	Hg:C	Hg:S	Hg Stock	CEC
		g kg^−1^			mg kg^−1^			mg m^−2^	cmol_(+)_ kg^−1^
Organic horizons (O)									
Fir	120	324 ± 81	25 ± 12	4.18 ± 0.34	0.458 ± 0.121	1.5 ± 0.4	215 ± 88	4.1 ± 1.7	67.7 ± 15.5
Pine	120	237 ± 145	13 ± 6	4.66 ± 0.44	0.143 ± 0.076	0.7 ± 0.2	147 ± 37	1.4 ± 0.7	48.4 ± 22.7
Mineral horizons (A) (0–10 cm)									
Fir	120	38 ± 19	0.4 ± 0.3	4.05 ± 0.24	0.117 ± 0.067	3.1 ± 0.1	348 ± 120	11.2 ± 5.0	16. 2 ± 6.5
Pine	120	25 ± 17	0.2 ± 0.1	4.70 ± 0.44	0.036 ± 0.021	1.6 ± 0.1	209 ± 65	4.0 ± 1.8	9.87 ± 5.1

Notes: *n*—number of samples, C—carbon, S—sulfur, Hg—mercury, Hg:C—mercury to carbon ratio, Hg:S—mercury to sulfur ratio, CEC—cation exchange capacity.

## Data Availability

The data presented in this study are available on request from the corresponding author.

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
