# Peer review of "Sequestration of Mercury in Soils under Scots Pine and Silver Fir Stands Located in the Proximity to a Roadway"

_ijerph, 2021, doi:10.3390/ijerph18094569_

Round 1

Reviewer 1 Report

  1. This paper was scanned with iThenticate resulting in 35% of similarity with another source in the internet. 8% or about 537 words are from the 1st author’s previous study. Please rewrite these parts to reduce the similarity rate. (scan result attached)
  2. This study suggest that roadway proximity does not show significant correlation with Hg concentration in soils and rather it has more effects from specific tree species. While this statement may be true based on the reported experiment, it can be easily misused to justify road pollution (which is a very significant contributor to air pollution in general). Therefore, the conclusion of this study should be presented very carefully by presenting a holistic view of the topic and the result. For example, in line 318 – 319, “in contrast with other reports), please cite and refer these other reports. Mention where exactly they differ and what could be the reason they differ).
  3. Finally, a disclaimer at the end of the study may be a wise move to avoid misinterpretation. For example some thing along the line of: “the result of this study …….. should not be interpreted as …… because…...”
  4. Limitation and further study avenue should also be boldly mentioned at the end of the study.

Author Response

#Reviever 1

This paper was scanned with iThenticate resulting in 35% of similarity with another source in the internet. 8% or about 537 words are from the 1st author’s previous study. Please rewrite these parts to reduce the similarity rate. (scan result attached).

Some parts of the text have been rewritten to avoid similarities, mainly in the methodology and discussion parts. There is also a matter of the writing style. We would like to stress that the most of the similar phrases was related to the similar methodology of previous article published by the first author (in previous work authors followed a similar approach of comparing study sites). Nevertheless, we made a number of changes to avoid this problem.

This study suggest that roadway proximity does not show significant correlation with Hg concentration in soils and rather it has more effects from specific tree species. While this statement may be true based on the reported experiment, it can be easily misused to justify road pollution (which is a very significant contributor to air pollution in general).

Yes, we agree, with this opinion, we expected much stronger pollution in the road vicinity, however the results do not encourage us to formulate such conclusion. Anyway, we tried to avoid the mentioned misuse of the results. This was done in a discussion section and conclusion 

Therefore, the conclusion of this study should be presented very carefully by presenting a holistic view of the topic and the result. For example, in line 318 – 319, “in contrast with other reports), please cite and refer these other reports.

Done

Mention where exactly they differ and what could be the reason they differ).

In this context, our data differ in the magnitude of the scatter of Hg to C relationship. Data from other reports needs to be log-transformed to show this relationship. We hope this has been clarified now. Concerning the reason of this scatter – we think that this can be due to the spatial variability of the samples. Our samples were taken from relatively small area,  while the mentioned reports present samples from large areas, with variable geology, forests stands,  altitude and soil type. But we also think that this can be stated base on our results, so we decided not to ad this explanation as a little speculative.  

Finally, a disclaimer at the end of the study may be a wise move to avoid misinterpretation. For example some thing along the line of: “the result of this study …….. should not be interpreted as …… because…...”Limitation and further study avenue should also be boldly mentioned at the end of the study.

These two comments were included as the end of conclusion section

Reviewer 2 Report

General comments:

The manuscript deals with "Fate of Hg in the soil under Scots pine and Silver fir stands". It is an interesting topic; however, the manuscript needs some revisions before considering for publication.

1. Page 1, Line 11; "The potential source of Hg pollution was a relatively new roadway traversing forested areas." In this study?? "In this study, the potential source of Hg pollution was a relatively new roadway traversing forested areas."

2. Write keywords alphabetically.

3. Quality of Figure 1 should be improved.

4. Table 2, Soil pH and CEC should be listed in the Table. pH and CEC have a great effect on fate of metals in the soil.

Author Response

#Reviever 2

General comments:

The manuscript deals with "Fate of Hg in the soil under Scots pine and Silver fir stands". It is an interesting topic; however, the manuscript needs some revisions before considering for publication.

  1. Page 1, Line 11; "The potential source of Hg pollution was a relatively new roadway traversing forested areas." In this study?? "In this study, the potential source of Hg pollution was a relatively new roadway traversing forested areas."

We emphasized that phrase according to the reviewer’s suggestion.

  1. Write keywords alphabetically.

Keywords were rewritten alphabetically.

  1. Quality of Figure 1 should be improved.

The quality of Figure 1 has been improved by uploading a picture in proper, better resolution.

  1. Table 2, Soil pH and CEC should be listed in the Table. pH and CEC have a great effect on fate of metals in the soil.

Soil pH is already listed in Table 2. We added the column containing CEC values. Also methods description were added

Round 2

Reviewer 1 Report

The paper has improved after revision

Reviewer 2 Report

Reviewers' comments have been addressed.